# Near-Haploidy and Low-Hypodiploidy in B-Cell Acute Lymphoblastic Leukemia: When Less Is Too Much

**DOI:** 10.3390/cancers14010032

**Published:** 2021-12-22

**Authors:** Oscar Molina, Alex Bataller, Namitha Thampi, Jordi Ribera, Isabel Granada, Pablo Velasco, José Luis Fuster, Pablo Menéndez

**Affiliations:** 1Josep Carreras Leukemia Research Institute, Campus Clinic, School of Medicine, University of Barcelona, 08036 Barcelona, Spain; ABATALLER@clinic.cat (A.B.); ntamphi@carrerasresearch.org (N.T.); 2Hematology Department, Hospital Clínic de Barcelona, University of Barcelona, 08036 Barcelona, Spain; 3Acute Lymphoblastic Leukemia Group, Josep Carreras Leukemia Research Institute, Campus ICO-Germans Trias i Pujol, Autonomous University of Barcelona, 08916 Badalona, Spain; jribera@carrerasresearch.org (J.R.); igranada@iconcologia.net (I.G.); 4Cytogenetics Laboratory, Hematology Department, Institut Català d’Oncologia (ICO), Hospital Germans Trias i Pujol, 08916 Badalona, Spain; 5Pediatric Oncology and Hematology Department, Hospital Vall d’Hebrón, 08035 Barcelona, Spain; pvelasco@vhebron.net; 6Pediatric Hematology and Oncology Department, Hospital Clínico Universitario Virgen de la Arrixaca, Instituto Murciano de Investigación Biosanitaria (IMIB), 30120 Murcia, Spain; josel.fuster@carm.es; 7Spanish Network for Advanced Therapies RICORS—TERAV, ISCIII, 28029 Madrid, Spain; 8Centro de Investigación Biomédica en Red de Cáncer (CIBER-ONC), ISCIII, 28029 Barcelona, Spain; 9Institució Catalana de Recerca i Estudis Avançats (ICREA), 08010 Barcelona, Spain

**Keywords:** hypodiploidy, near-haploidy, B-cell acute lymphoblastic leukemia, clinical biomarkers, patient stratification

## Abstract

**Simple Summary:**

B-cell acute lymphoblastic leukemia (B-ALL) is characterized by an uncontrolled proliferation of blood cells in the bone marrow. A small fraction of B-ALL patients shows abnormally low chromosome numbers, defined as hypodiploidy, in leukemic cells. Hypodiploidy with less than 40 chromosomes is a rare genetic abnormality in B-ALL and is associated to an extremely poor outcome, with low survival rates both in pediatric and adult cases. In this review, we describe the main clinical and genetic features of hypodiploid B-ALL subtypes with less than 40 chromosomes, the current treatment protocols and their clinical outcomes. Additionally, we discuss the potential cellular mechanisms involved on the origin of hypodiploidy, as well as its leukemogenic impact. Studies aiming to decipher the biological mechanisms involved in hypodiploid subtypes of B-ALL with less than 40 chromosomes are crucial to improve the poor survival rates in these patients.

**Abstract:**

Hypodiploidy with less than 40 chromosomes is a rare genetic abnormality in B-cell acute lymphoblastic leukemia (B-ALL). This condition can be classified based on modal chromosome number as low-hypodiploidy (30–39 chromosomes) and near-haploidy (24–29 chromosomes), with unique cytogenetic and mutational landscapes. Hypodiploid B-ALL with <40 chromosomes has an extremely poor outcome, with 5-year overall survival rates below 50% and 20% in childhood and adult B-ALL, respectively. Accordingly, this genetic feature represents an adverse prognostic factor in B-ALL and is associated with early relapse and therapy refractoriness. Notably, half of all patients with hypodiploid B-ALL with <40 chromosomes cases ultimately exhibit chromosome doubling of the hypodiploid clone, resulting in clones with 50–78 chromosomes. Doubled clones are often the major clones at diagnosis, leading to “masked hypodiploidy”, which is clinically challenging as patients can be erroneously classified as hyperdiploid B-ALL. Here, we summarize the main cytogenetic and molecular features of hypodiploid B-ALL subtypes, and provide a brief overview of the diagnostic methods, standard-of-care treatments and overall clinical outcome. Finally, we discuss molecular mechanisms that may underlie the origin and leukemogenic impact of hypodiploidy and may open new therapeutic avenues to improve survival rates in these patients.

## 1. Introduction

Acute lymphoblastic leukemia (ALL) is a neoplasm arising from lymphoid precursor cells and can be classified as B-ALL or T-ALL based on the immunophenotype of the neoplastic cells [1]. The global incidence of ALL is ~3 cases per 100,000 people and shows a bimodal distribution, with a predominant peak early in life (1 to 15 years) and a second, much lower, peak in older groups (>55 years) [2] (Figure 1). ALL has a slightly higher incidence in males, with a male-to-female ratio of 1.2:1 [3]. The disease is characterized by the uncontrolled proliferation of leukemic cells, which invade the bone marrow (BM), peripheral blood (PB), and other hematopoietic tissues including spleen, liver, and lymph nodes, resulting in a hematopoietic displacement which is responsible for the cytopenias frequently observed at diagnosis. ALL cells also infiltrate commonly the central nervous system (CNS).

B-cell precursor ALL (B-ALL) accounts for 80–85% of ALL cases and is characterized by small-medium sized leukemic blast cells staining almost always positive for the B-cell antigens CD19, cytoplasmic CD79a and CD22. Although BM and PB are involved in most cases, B-ALL occasionally presents with primary nodal or extranodal sites (B-lymphoblastic lymphoma), which predominantly affect skin, soft tissue, bone and lymph nodes [4]. Currently, B-ALL is classified by the World Health Organization as B-lymphoblastic leukemia/lymphoma with recurrent genetic abnormalities (Table 1) and B-lymphoblastic leukemia/lymphoma not otherwise specified. Among the B-ALL subtypes with recurrent genetic abnormalities, B-ALL with hypoploidy is classified as a well distinguished entity whose blasts contain <46 chromosomes (see below) [1]. 

There has been an extraordinary improvement in outcomes for B-ALL during the last four decades, which have been more pronounced in children than in adults (5-year overall survival [OS] of ~90% compared with ~25% for patients >50 years) [5,6]. There are several known risk factors that can help to stratify patients with B-ALL, including adverse demographic factors such as age <1 or ≥10 years [7,8]; and also adverse clinical features at diagnosis, including CNS infiltration or high white blood cell (WBC) count (>50 × 10^9^/L) [9]. Genetics and cytogenetics also have a direct impact on prognosis [10,11]. Among other genetic features, aneuploidy—defined as the gain or loss of one or more whole chromosomes—is an important prognostic factor [12], and will be examined in this review; specifically, the favorable risk of hyperdiploidy and the adverse risk of hypoploidy. Other genetic findings with prognostic impact used to stratify patients in treatment protocols have been recently discussed elsewhere [13,14]. Finally, one important risk factor with prognostic value is the response to treatment, evaluated as the detection of minimal residual disease (MRD) at specific timepoints of treatment [15].

## 2. Definition of Hypodiploid B-ALL Subgroups

Hypodiploidy -the loss of one or more whole chromosomes- is a rare cytogenetic finding (≤7%) in children and adults with B-ALL and is generally an adverse prognostic marker [12,16,17,18,19,20,21,22,23,24,25,26]. Most cases (~80%) of hypodiploid B-ALL present with 45 chromosomes and are classified as near-diploid B-ALL, a clinically distinct entity characterized by rearrangements that form dicentric chromosomes but that does not have outcomes as poor as those associated with hypodiploid B-ALL [21]. Hypodiploid B-ALL is strictly defined by most studies as ≤44 chromosomes and can be further subdivided based on chromosome number as: (i) high-hypodiploid B-ALL (40–44 chromosomes), (ii) low-hypodiploid (30–39 chromosomes), and (iii) near-haploid B-ALL (24–29 chromosomes) [17,20,24,26]. Additionally, a DNA index (determined by flow-cytometry) below 0.65 and between 0.65 to 0.82 may also be used to identify near-haploidy and low-hypodiploidy, respectively, in B-ALL samples [24]. The karyotypes and clinical outcomes of high-hypodiploid B-ALL cases differ from those of near-haploid and low-hypodiploid B-ALL, which display significantly poorer clinical outcomes across all age groups [21]. In the present review, we will focus on B-ALL with hypodiploidy of <40 chromosomes—near-haploidy and low-hypodiploidy subgroups—considered as separate cytogenetic entities associated with a very poor prognosis [27]. 

### 2.1. Demographic Features of B-ALL with Hypodiploidy <40 Chromosomes

Hypodiploidy is very rare in both childhood and adult B-ALL [27]. Near-haploidy is restricted to childhood B-ALL and represents ~0.5% of all B-ALL cases [21,28], while patients with low-hypodiploidy include both children and adults with frequencies of ~0.5% and ~4%, respectively [21,24,29]. Within the pediatric population, patients with near-haploidy present at a younger age (median age at diagnosis of 6.2 years), whereas low-hypodiploidy is more common in older children, with a median age at diagnosis of 12.9 years [17,21,23,24] (Table 2). Several studies have reported a higher proportion of males in the near-haploid and low-hypodiploid groups [22,24,26]. For example, in a retrospective analysis by the Ponte di Legno Childhood ALL Working Group (PDLWG), the male-to-female ratio was 1.46 in 101 cases with near-haploidy and 1.14 in 118 cases with low-hypodiploid B-ALL [24]. However, male predominance is not consistently reported in different studies [17,21] (Table 2).

### 2.2. Clinical and Biological Features of B-ALL with Hypodiploidy <40 Chromosomes

Generally, patients with hypodiploid B-ALL present with a lower diagnostic WBC than patients with non-hypodiploid B-ALL [16]; however, some studies have reported a higher WBC in the near-haploid group [11,21,24]. The aforementioned PDLWG study [24] found a median WBC at presentation of 21.8 × 10^9^/L in the near-haploid group against 7 × 10^9^/L in the low-hypodiploid group. In addition, up to 49% of patients with a near-haploid karyotype presented with ≤20 × 10^9^/L compared with 85% in the low-hypodiploid group, whereas 31% in the near-haploid group presented with >50 × 10^9^/L WBC compared with only 4% in the low-hypodiploid group. These findings are in line with previous reports [21,22,24] (Table 2).

Regarding cell morphology, French-American-British (FAB) L2/L1 or L2 subtypes were reported to be more frequent at presentation in patients with hypodiploid B-ALL. Indeed, up to 50% of near-haploid cases were non-L1 FAB subtype and 25% were L2 subtype in a series reported by the Children Cancer Group (CCG) [16,19]. Patients with B-ALL with hypodiploidy show a B-cell precursor immunophenotype, with positivity for CD19, CD34, CD22, cCD79a and TdT [22,27]. Near-haploid ALL is generally also positive for CD10, whereas low-hypodiploid ALL may exhibit a more immature B-cell precursor immunophenotype, lacking CD10 in a substantial proportion of cases, consistent with a pro-B ALL phenotype [30]. Additionally, a very small proportion of children with hypodiploid ALL present with a T-ALL immunophenotype, usually with positivity for TdT and expressing the T-cell specific markets CD1a, CD2, CD3, CD5, and CD7 [27,31,32]. In a larger series presented by the PDLWG, Nachman et al. and the UK Medical Research Council, only 1% of the total cases with hypodiploidy <40 chromosomes showed a T-cell immunophenotype (Table 2), whereas 11–15% of high-hypodiploid cases exhibited a T-ALL immunophenotype [21,22,24].

### 2.3. Extramedullary Involvement at Presentation

CNS involvement, which is defined as the presence of blasts in a sample of cerebrospinal fluid with ≥5 leukocytes/µL and <10 erythrocytes/µL or the presence of cranial nerve palsies (CNS3) [33], is rare at diagnosis and was recorded in only one out of 115 and 97 cases (1%) of near-haploid and low-hypodiploid ALL, respectively, in the PDLWG study [24,26]. Other frequent leukemic extramedullary involvements at diagnosis of near-haploid and low-hypodiploid B-ALL is at the spleen and liver, presenting as splenomegaly in 44% and 28% of cases, and hepatomegaly in 43% and 35%, respectively [24,26]. By contrast, mediastinal mass and lymphadenopathy is less frequent, with 7% and 13% and 30% and 14% of cases with near-haploid ALL and low-hypodiploid ALL, respectively [24]. Testicular disease at diagnosis is also rare in both types, with no reported cases in the AALL03B1 study and only 2 of 46 cases (4%) of near-haploid B-ALL in the PDLWG study [24]. Taking these findings together, it seems clear that most of the clinical and biological features in patients with near-haploid and low-hypodiploid B-ALL do not seem to be predictive of poor outcome.

## 3. Cytogenetic Characterization of B-ALL with Hypodiploidy <40 Chromosomes

Although traditionally defined as the loss of one or more chromosomes in a cell [34], hypodiploidy has been classified according to diverse criteria in different studies to define different hypodiploid B-ALL subtypes (Table 3), and a clear distinction between those cases with <40 chromosomes and those with 40–45 chromosomes has become generally accepted based on the evident differences in treatment response [21,35]. Hypodiploid cases with <40 chromosomes can be further subdivided into two groups based on the bimodal distribution of chromosome numbers: (i) near-haploidy, with 24–29 chromosomes (Figure 2A), and (ii) low-hypodiploidy, with 30–39 chromosomes (Figure 2B,C) [27] (Table 3). Although the modal number of chromosomes is variable, the most recurrent modal numbers are 25–28 for near-haploid and 33–39 for low-hypodiploid B-ALL [36]. Based on conventional chromosome banding analyses, hypodiploidy with <40 chromosomes shows a non-random loss of chromosomes: in near-haploid B-ALL, retained disomies generally comprise chromosomes 8, 10, 14, 18, 21, X and Y [21,22,30,37] whereas in low-hypodiploid cases, retained disomies are more variable and typically comprise chromosomes 1, 5, 6, 8, 10, 11, 14, 18, 19, 21, 22, X and Y, with retained disomies for chromosomes 1, 6, 11 and 18 being the most frequently observed. The most typically lost chromosomes are chromosome 3, 7, 9, 15, 16 and 17 [21,22,30,38,39] (Table 3). The non-random retention of chromosomes suggests that these chromosomes may harbor specific genes that enhance the oncogenic potential of leukemic cells. 

Structural chromosome aberrations (i.e., chromosomal translocations) in hypodiploidy of <40 chromosomes are extremely rare, especially in near-haploid B-ALL [21,22,38,42], suggesting that massive chromosomal loss alone may be sufficient for leukemogenesis. Patients with low-hypodiploid B-ALL show some additional alterations, albeit few, in the karyotype beyond monosomies (Figure 2B,C). It has been reported that pediatric patients with B-ALL with <36 chromosomes may show additional structural chromosome aberrations more frequently when compared with those with ≥36 [22]. It would be interesting to learn whether pediatric patients with additional structural alterations are also older than those without them, which would be consistent with different reports on adult B-ALL showing structural chromosome aberrations, especially unbalanced translocations and losses of chromosome arms, in 50% of low-hypodiploid cases [35,38]. 

### “Masked Hypodiploidy”: A Clinical Challenge

A feature common to patients with near-haploid and low-hypodiploid B-ALL is the presence of a clone with an exact or near-exact chromosome doubling of the hypodiploid clone, resulting in a clone with a modal chromosome number of 50–78, in the high-hyperdiploid or triploid range [17,21,22,41] (Table 3) (Figure 2A). Notably, some chromosomes are preferentially lost after chromosome doubling; typically, chromosomes 2, 5, 6, 10, 14 and 22 [38]. The presence of hypodiploid doubled clones has been observed in ~60–65% of patients with near-haploid and low-hypodiploid B-ALL in different studies, and is commonly observed as a mosaic with both hypodiploid and hyperdiploid (doubled) clones visible by standard cytogenetics, fluorescence in situ hybridization (FISH) or flow cytometry analysis of DNA content [21,41,43]. Furthermore, the doubled clone may be the only one detected at diagnosis, leading to the manifestation known as “masked hypodiploidy”, which is clinically challenging since patients can be erroneously classified and treated for high-hyperdiploid B-ALL while being at higher risk of treatment failure. It has been reported that there is no difference in clinical outcome for patients with “masked hypodiploidy”, those who are mosaic for a doubled clone and a hypodiploid clone, and those who have only a hypodiploid clone [22,35,41]. In addition, the hypodiploid clone tends to be quantitatively more frequent at relapse, suggesting that the actual hypodiploid clones may be more chemoresistant than their hyperdiploid (doubled) counterparts [20,44].

There is evidence to suggest that chromosomal doubling arises by endoreduplication [17]: for instance, (i) the hyperdiploid clone is an exact duplication of the near-haploid or low-hypodiploid clone; (ii) in rare cases, the structural chromosomal abnormalities are also shared in paired chromosomes; and (iii) hyperdiploid cell lines have been established in culture from near-haploid B-ALL patient samples [45,46]. However, studies aiming to elucidate the molecular mechanism(s) leading to the chromosomal doubling have not been performed to date. 

Whole-genome doubling (WGD) is a common phenomenon in nature that generates cells referred to as “polyploid”, containing multiple copies of the complete set of chromosomes [47]. Indeed, polyploidy has been reported to be indispensable for normal development and organ formation across various organisms, from fungi to humans [47,48], and has more recently been associated with wound healing and tissue homeostasis [48]. Additionally, polyploidization has been implicated in early carcinogenesis and neoplastic progression, as polyploid cells have been reported to be more permissive to aneuploidy through their ability to buffer deleterious mutations that affect cellular fitness [49,50]. In this line, it is tempting to speculate that WGD is even more relevant in near-haploid and low-hypodiploid cellular backgrounds, as it generates chromosomally-doubled clones with presumably lower cell-fitness costs and higher adaptation capacity than their hypodiploid counterparts. WGD arises as a consequence of alternative cell cycle programs referred to as endoreduplication or endocycles, through which cells successively duplicate genomic DNA without segregating their chromosomes during mitosis [47]. Only the most extreme abbreviations of the mitotic phase are correctly referred to as endoreduplication. The cell cycles that include some mitotic processes, such as chromosome condensation, nuclear envelope breakdown and spindle formation, are referred to as “endomitosis”, which abort mitosis mostly during metaphase or anaphase and lack cytokinesis [47]. These cases are associated with mitotic defects leading to mitotic slippage. In both scenarios, the result is either a cell that maintains separate nuclei, generating multinucleate cells, or a cell with an enlarged single nucleus. Endoreduplication cycles use much of the same machinery that regulates the transition from G1 to S phase in normal “mitotic cell cycles”. To convert the mitotic cell cycle into an endoreduplication cycle, the cell cycle must be altered in two essential ways. On the one hand, by bypassing the central processes of mitosis, chromosome segregation and cytokinesis, without blocking DNA replication. This is accomplished in plants and animal cells by downregulating the cyclin-dependent kinases (CDKs) that drive G2-to-M phase progression, while allowing continuous activity of the CDKs that drive G1-to-S progression. On the other hand, periodic inactivation of CDKs involved in G1-to-S phase transition to enable a G1-like gap with low S-phase-specific CDK activity during which pre-replication complexes can be reassembled [47,48]. The precise mechanism underlying WGD in near-haploid and low-hypodiploid B-ALL is currently not known, but as this phenomenon is very common and exclusive of these subtypes of B-ALL, it could be hypothesized that it has an important role on the development of these subtypes of ALL. 

## 4. Molecular Characterization of Hypodiploid B-ALL with <40 Chromosomes

In addition to the massive genetic losses, both near-haploid and low-hypodiploid B-ALL show characteristic and differentiated gene expression profiles, in addition to specific mutational and focal copy-number alteration (CNA) landscapes, those excluding whole chromosome losses [30]. Notably, near-haploid and low-hypodiploid B-ALL presenting with or without doubled clones show similar transcriptional and mutational profiles [30], most likely explaining the similar clinical outcomes between patients with and without chromosomal doubling [17,22,41]. 

### 4.1. Near-Haploid B-ALL 

The mutational landscape of near-haploid B-ALL is characterized mainly by the presence of alterations involving receptor tyrosine kinases and activating RAS signaling alterations, with >70% of patients showing mutations or focal CNA involving genes in these pathways (Table 4) [30,38,51]. The different RAS signaling alterations have been shown to be mutually exclusive, suggesting that, in contrast to the convergent evolution for RAS mutations observed in infants with MLL-rearranged B-ALL [52], a single alteration in the pathway is sufficient to maintain constitutive RAS-pathway activation. Focal deletions or point mutations in *NF1* gene are the most recurrent genetic alterations of near-haploid B-ALL (≥44% of patients) [23,30,53]. *NF1* encodes a RAS–GTPase-activating protein that negatively regulates Ras signaling. Deletions affecting this gene are intragenic and involve exons 15–35 in the majority of cases, most likely due to illegitimate recombinase activating gene (RAG) activity [30]. Other recurrently mutated RAS pathway genes in near-haploid B-ALL are *NRAS* (15% of patients), *FLT3* (9% of patients), *KRAS* (3% of patients) and *PTPN11* (1.5% of patients) [30,38,51]. Some of these mutations have been described in patients diagnosed with Noonan syndrome, such as the *PTPN11* Gly503Arg substitution, and have been also described in remission samples or in T-cell lymphocytes, such as *NRAS* Gly12Ser [30], suggesting that the single-nucleotide variants (SNVs) are germline SNVs and that near-haploidy may be a manifestation of this cancer-predisposing syndrome in some instances. Additionally, alterations in *PAG1* (mostly deletions) have been reported in 10% of patients with near-haploid B-ALL [23,30]. Most of the deletions were homozygous and affected the upstream region and first exon of the gene, leading to a complete loss of gene expression [30]. *PAG1* encodes for a transmembrane adaptor protein that binds to the tyrosine kinase CSK protein, which negatively regulates SRC-family kinases involved in both Ras and B-cell receptor signaling pathways. Importantly, *PAG1* deletions together with positive MRD status have been associated with poorer outcomes and increased incidence of relapse in hypodiploid ALL [30]. Other recurrent focal CNAs in near-haploid B-ALL include *CDKN2A/B* at 9p21.3 (22% of patients), a histone cluster at 6p22 (19% of patients), *IKZF3/Aiolos* at 17q12 (13% of patients), *RB1* at 13q14.2 (9% of patients) and *PAX5* at 9p13.2 (7% of patients) [30]. As with other B-ALL subtypes, patients with near-haploid B-ALL have recurrent disruption, by deletion, insertion-deletions or point mutations, of the lysine acetyltransferase gene *CREBBP* (32% of patients) [30]. CREBBP mediates the expression of glucocorticoid-responsive genes and may play an active role in response to glucocorticoids, suggesting that patients harboring *CREBBP* alterations may be more prone to therapy failure and relapse [54]. Point mutations in other histone-modifier genes, such as *EP300* and *EZH2*, are also observed in near-haploid B-ALL albeit at lower frequencies (<5% of patients). Of note, alterations involving epigenetic modifiers are more frequently observed in relapsed B-ALL than at diagnosis, highlighting their role in leukemia progression and clonal evolution. 

### 4.2. Low-Hypodiploid B-ALL

The genetic hallmark of low-hypodiploid B-ALL is *TP53* mutations, which are observed in >90% of patients in both childhood and adult low-hypodiploid B-ALL (Table 4) [30,38,51,55]. Most are missense mutations in exons 5–8, affecting the DNA-binding domain and the nuclear localization sequence [30,38]. Other characteristic and recurrent genetic alterations in the low-hypodiploid B-ALL subtype are *RB1* mutations or deletions (41% of cases), deletions of *IKZF2/Helios* (53% of cases) and deletions of *CDKN2A/B* genes (22% of cases) [30,38]. Mutations of *TP53* are found in homozygosity in virtually all low-hypodiploid B-ALL cases due to the very recurrent loss of chromosome 17. *TP53* mutations are frequently found in non-tumor hematopoietic cells in 50% of the cases of childhood low-hypodiploid B-ALL [38,51], suggesting that these cases may be a manifestation of Li-Fraumeni syndrome or other germline *TP53* cancer-predisposing mutations [30,55,56]. Accordingly, genetic counseling is recommended for children with low-hypodiploid B-ALL carrying *TP53* mutations, and their relatives [57,58]. In contrast to childhood cases, *TP53* mutations in low-hypodiploid adult B-ALL are somatic, are not found in healthy hematopoietic cells, and not detectable in remission samples [30,38]. 

Focal deletions involving lymphocyte development genes, such as *IKZF1*, *EBF1* or *LEF1*, are not particularly frequent in low-hypodiploid B-ALL samples. Instead, *IKZF2*/*Helios* loss at 2q34 is the most recurrent alteration involving genes associated with B-cell differentiation (53% in children and 36% in adults), being much more frequently observed in low-hypodiploid B-ALL than in any other B-ALL subtype, including near-haploid ALL (Table 4). *IKZF2/Helios* is highly expressed in common lymphoid progenitors and in pre-pro-B progenitor cells, which is consistent with the immunophenotype observed in low-hypodiploid B-ALL of lower frequencies of antigen-receptor rearrangements, CD19 and CD10 [30,59]. These findings reinforce the hypothesis that low-hypodiploidy targets a more immature lymphoid progenitor (pro-B stage or earlier) than other B-ALL subtypes with a pre-B immunophenotype. Indeed, in one study patients with *IKZF2/Helios* deletions had a poorer outcome than patients with wild-type *IKZF2/Helios* in an univariate analysis [30]. Beyond lymphocyte development, alterations of cell cycle regulators are also frequently observed in both childhood and adult low-hypodiploid B-ALL. As discussed earlier, alterations in *RB1* are observed in 41% and 19% of childhood and adult patients, respectively. Additionally, *CDKN2A/B* mutations are observed in 22% of both childhood and adult low-hypodiploid B-ALL samples, and are often mutually exclusive with *RB1* mutations [58]. The finding that *TP53* is ubiquitously perturbed in low-hypodiploid B-ALL suggests that mutations in an additional member of the TP53/RB1 pathway (either *RB1* or *CDKN2A/B*) would be sufficient to deregulate cell proliferation in these cases. Furthermore, mutations in histone-modifier genes, such as *CREBBP*, are detected in 60% of patients with low-hypodiploid B-ALL [30]. Strikingly, while patients with low-hypodiploid B-ALL do not show recurrent mutations in genes associated with RAS pathway activation, as do patients with near-haploid B-ALL, transcriptomic analyses of pathway activation in low-hypodiploid B-ALL revealed constitutive signaling of *RAS* and *PI3K* pathways [30]. 

A recent study by the Japan Association Childhood Leukemia Study Group (JACLS) reported a high frequency of mutations in *CIC* in both low-hypodiploid and near-haploid B-ALL, present in 5 of 9 patients [53]. *CIC* is a member of the high mobility group (HMG)-box superfamily of transcriptional repressors and is recurrently mutated in oligodendrogliomas and in round cell sarcomas. However, the germline status of *CIC* in B-ALL with hypodiploidy of <40 chromosomes is unknown and raises the question of whether it could be more prevalent in Asian ancestors than in non-Asian populations [53].

### 4.3. Proper Identification of Hypodiploid B-ALL with <40 Chromosomes

As previously mentioned, “masked hypodiploidy” represents an important diagnostic challenge, as it may be mis-diagnosed with high-hyperdiploidy of 51–65 chromosomes (Figure 2A). Conventional cytogenetic G-banding analysis provides the modal number of chromosomes, which is essential for the differentiation between these cytogenetic groups with <40 chromosomes, high-hyperdiploid B-ALL and, especially, for those B-ALL cases presenting with >65 chromosomes (near-triploidy), which should always be considered as doubled clones from a low-hypodiploid B-ALL. Cases between 51 and 65 chromosomes, in the range of the high-hyperdiploidy and masked near-haploidy, are especially difficult to define and a detailed analysis of the gained chromosomes is crucial to distinguish between the two B-ALL entities. The major cytogenetic characteristic of masked near-haploid B-ALL is the presence of mainly tetrasomies, which is in contrast to high-hyperdiploidy, which mainly shows trisomies (with the exception of chromosome 21). Accordingly, the presence of tetrasomies other than tetrasomy of chromosome 21 might suggest a masked hypodiploid clone. Moreover, the subsequent loss of some chromosomes is very frequently observed after chromosome doubling in cases of B-ALL with hypodiploidy of <40 chromosomes, suggesting certain levels of chromosome instability and, therefore, not all gained chromosomes are tetrasomic in the karyotypes of masked near-haploid samples [38]. 

Complementary techniques may be used when there is a suspicion of masked hypodiploidy, including interphase FISH or flow-cytometry analyses of DNA index. These techniques allow scoring a higher number of cells in the samples and may facilitate the detection of hypodiploid clones that were not detectable by conventional cytogenetics. Likewise, single nucleotide polymorphism (SNP)-arrays have emerged as a very reliable tool to identify masked hypodiploid clones, especially their capacity to detect loss of heterozygosity (LOH). Indeed, the presence of LOH in most chromosomes, especially those typically lost in the original hypodiploid clones (Table 3), is defining the near-haploid and low-hypodiploid entities. Some authors have recently provided simple and helpful algorithms for proper discrimination between masked hypodiploidy from high-hyperdiploid B-ALL using SNP-arrays and focusing on specific chromosome gains, such as chromosomes 1, 7 and 14 (Figure 2D) [39].

## 5. Etiology of Hypodiploidy in B-ALL

No preclinical models of near-haploid or low-hypodiploid B-ALL subtypes are currently available, preventing the direct study of the biological causes and the pathogenic consequences of hypodiploidy in B-ALL. Indeed, most studies describing hypodiploid subtypes in B-ALL are clinical reports, patient sample analyses and retrospective analyses of clinical outcomes. Accordingly, there is only circumstantial evidence for the cellular mechanisms leading to hypodiploidy and its pathogenic consequences. Genomic analyses of these subtypes have been difficult given the limited number of cases; however, a study on a small cohort of 8 near-haploid and 4 low-hypodiploid B-ALL samples suggested that the massive loss of chromosomes is the primary oncogenic event, with other oncogenic insults occurring after hypodiploidy [37]. This is consistent with similar analyses in high-hyperdiploid B-ALL cases, the most frequent aneuploid entity in B-ALL, indicating that chromosome gains were the primary oncogenic event [60,61]. Thus, similar pathogenic mechanisms involving gross aneuploidies may be shared in these B-ALL subtypes. Furthermore, the genomic landscape of near-haploid and low-hypodiploid B-ALL subtypes, as well as that of high-hyperdiploid subtypes, is characterized by aneuploidy and subtype-specific mutations (see above), with significant fewer microdeletions and structural chromosomal rearrangements in comparison with other cytogenetic subtypes containing structural chromosomal reorganizations [30,60]. Collectively, these data strongly suggest that hypodiploidy has a direct impact on cell transformation and leukemogenesis rather than being solely a passenger event. The fact that severe hypodiploidy is observed in a wide spectrum of neoplasms further indicates that it is indeed a major contributor of tumorigenesis [62]. 

Although the functional impact of extreme hypodiploidy on leukemia initiation is poorly understood, a common assumption is that the widespread LOH resulting from hypodiploidy leads to the unmasking of recessive alleles or to gene-dosage imbalances affecting oncogenes, tumor-suppressor genes, and also protein imbalances affecting different aspects of cell physiology, proliferation and/or survival [30,37,51]. Indeed, loss-of-function mutations of *TP53* are a hallmark of both childhood and adult low-hypodiploid B-ALL [30], suggesting that alterations in this gene are an important event in the pathogenesis of this B-ALL subtype. Alternative mechanisms for tumor suppressor pathway inactivation may be involved in the pathogenesis of near-haploid B-ALL; for instance, microdeletions of *CDKN2A/B* loci on chromosome 9, typically monosomic in near-haploid B-ALL, are common in this hypodiploid subtype [30]. These proteins are involved in the control of the tumor-suppressor *RB1* and their inactivation leads to a dysregulation of the G1-to-S checkpoint leading to uncontrolled proliferation [30,51]. 

Remarkably, the pattern of chromosomal retention in near-haploid B-ALL is similar to the pattern of commonly gained chromosomes in the high-hyperdiploid group [17,30]. Thus, retention of both homologs of specific chromosomes, especially 14, 18, 21 and X, may play an important role in leukemogenesis in these cases. Elucidating the role and cooperativity of the genes on these chromosomes, and their contribution to different cellular pathways, should shed new light on the pathogenic mechanisms resulting from hypodiploidy in B-ALL. In particular, the contribution of alterations in the RAS signaling pathway, Ikaros-family alterations and/or *TP53* alterations in leukemogenesis are a subject of active current investigation [58]. 

## 6. Origin of Near-Haploidy and Low-Hypodiploidy 

While the near-haploid and low-hypodiploid chromosomal patterns have been recognized for many years [16], the factors driving the generation of these aneuploidies and their associated genomic alterations remain poorly understood. They may originate through successive loss of chromosomes or by a single erroneous mitosis event in an early hematopoietic stem/progenitor cell. The latter hypothesis is currently the most accepted owing to the bimodal distribution of chromosome numbers reported in the literature for these hypodiploid B-ALL subtypes [37]. Multipolar mitosis or an abnormal partial mitotic pairing of homologous chromosomes appear to be the most feasible mechanisms underlying hypodiploidy [17,62], but no experimental evidence has been provided to support this hypothesis. Remarkably, high-resolution microscopic analyses of dividing primary hypodiploid B-ALL cells stained with antibodies that recognize different cytoskeleton structures revealed an increase in the frequency of multipolar spindles when compared with non-hypodiploid B-ALL cells (data not previously published) [63], suggesting that multipolar spindles could indeed be a crucial cellular mechanism leading to hypodiploidy. However, whether these multipolar spindles are the cause or a consequence of the chromosome losses remains an open question. 

Given that the immunophenotype of both near-haploid and low hypodiploid B-ALL is consistent with that of a common pre-B or pro-B leukemia, respectively [17,30], it is likely that hypodiploidy arises in a similar hematopoietic progenitor as most B-ALLs. Notably, pre-leukemic clones with different chromosome rearrangements and high-hyperdiploidy have been observed in cord blood samples and in neonatal heel prick tests from patients that later develop childhood B-ALL [64]. This finding suggests that the primary genetic abnormalities arise in utero during fetal hematopoiesis and act as pre-leukemic initiating events that remain clinically silent upon the acquisition of secondary cooperating genetic alterations, which are necessary to promote leukemia development [65]. Of note, although no direct evidence has been provided for near-haploid and low-hypodiploid B-ALL subtypes, it is tempting to speculate that hypodiploidy in childhood B-ALL also arises in utero as a consequence of an aberrant mitosis event in early hematopoietic progenitors. The pathogenic mechanisms of extreme hypodiploidy and their contribution to leukemogenesis are the subject of investigation, as the identification of novel therapeutic targets are crucial for developing new treatments to improve the dismal outcome of these rare subtypes of B-ALL. 

## 7. Outcome and Treatment Strategies for B-ALL with Hypodiploidies <40 Chromosomes

### 7.1. Event-Free Survival and Overall Survival Rates

Patients with hypodiploid B-ALL have an overall very poor clinical outcome, particularly in those cases with <40 chromosomes [17]. While the survival rates are increasingly dismal with decreasing modal chromosome numbers, the outcome of patients with either near-haploidy and low-hypodiploidy do not significantly differ, being similarly poor in both groups with modal numbers below 40 chromosomes [20,21,24]. The poor prognostic impact conferred by hypodiploidy with <40 chromosomes is maintained in multivariate analyses after adjustment for important risk factors including age, WBC count, and Philadelphia/t(9;22) status [19]. Indeed, the MRC UK-ALL trials of childhood B-ALL cases reported 3-year event-free survival (EFS) rates of 29% versus 65% in hypodiploidy below and above 40 chromosomes, respectively [21]. Its prognosis in adults remains extremely poor, with 5-year EFS rates ~20%, despite stratification by high-risk treatment protocols [27]. Importantly, the poor outcome and low EFS rates of B-ALL with hypodiploidy with <40 chromosomes has been consistently reported by different studies (Table 5). 

The low EFS rates observed in near-haploid and low-hypodiploid groups are related to a high relapse rate mostly from isolated BM relapses, and succumbing to the disease after first remission [24,25] (Table 5). According to the review by Groeneveld-Krentz et al. on patients with relapsed BCP-ALL [68], those with pediatric B-ALL with hypodiploidy <40 chromosomes show specific characteristics compared with other B-ALL groups, including: (i) association with older age at initial diagnosis, (ii) shorter time to first relapse, (iii) more frequent allocation to the high-risk treatment arm of the relapse trial, and (iv) inferior second remission rates [68]. Remarkably, patients with hypodiploidy with <40 chromosomes presenting with a predominant doubled clone or masked hypodiploidy had late relapses and more often achieved a second remission, as compared with patients with a predominant or exclusive hypodiploid clone. Despite these differences, however, the outcome of these patients is similarly poor, highlighting that all hypodiploidies with <40 chromosomes should be classified as high-risk irrespective of time to relapse [68]. 

### 7.2. Relationship of Genetic and Clinical Features with Patient Outcome

The EFS is not significantly different between patients with near-haploid or low-hypodiploid B-ALL, including those cases with “masked hypodiploidy” [23,24,57]. In some cases, hypodiploidy may accompany other primary genetic abnormalities, such as *BCR-ABL1*, *TCF3-PBX1*, *ETV6-RUNX1* and *KMT2A* rearrangements, which modulate the prognosis of the disease. Accordingly, some authors have suggested that these patients should be treated based on the primary structural abnormalities rather than the hypodiploidy, and on their MRD values after induction [24]. The high presence of germline *TP53* mutations among patients with low-hypodiploidy confer an increased risk of relapse in this group and is associated with the development of secondary neoplasms [24]. Therefore, it is highly recommended that all patients with low-hypodiploidy B-ALL are tested for germline *TP53* mutations [24,69]. Strikingly, the germline *TP53* mutations in these cases have been associated with increased mortality due to second neoplastic malignancies following hematopoietic stem cell transplantation (HSCT), highlighting the importance of the germline study in low-hypodiploid B-ALL to assess HSCT versus less toxic alternative therapies [26,67,70]. The presence of other recurrent mutations in near-haploid and low-hypodiploid B-ALL cases, such as alterations in the RAS-pathway, *IKZF* or *RB1*, has not shown a clear association with patient prognosis [23]. However, the small sample sizes, owing to the rarity of these B-ALL subtypes, may mask a significant association of these mutations with patient outcomes. Interestingly, a study by The Children’s Healthcare of Atlanta found that patients with hypodiploid B-ALL with <40 chromosomes and age ≥10 years display significant worse 2-year EFS rates (33.3% vs. 100%) [57]. Noteworthy, the impact of age on prognosis has also been reflected in other studies of adult B-ALL [35], where low-hypodiploid B-ALL patients < 35 years had survival rates of 71% against 21% in those patients >35 years.

The most important prognostic factor for near-haploid and low-hypodiploid B-ALL groups is the MRD status at the end of induction (EOI) [23]. Univariateanalyses including different clinical factors, such as National Cancer Institute (NCI) risk groups, MRD EOI status or HSCT, showed that only MRD EOI <0.01% was a significant prognostic factor for EFS in these cases [26]. Despite the poor prognosis of hypodiploid B-ALL with <40 chromosomes, the outcome improves significantly if the MRD EOI is negative, with 5-year EFS of 68–85% vs. 44–50% for MRD negative and positive patients, respectively [22,23,26,57,67] (Table 5). In view of these data, several groups have intensified treatments based on MRD-based stratification protocols, which has proven to be the most appropriate strategy associated with more favorable outcomes [24].

### 7.3. Current Treatment Protocols

Different study groups, such as the UKALL, NOPHO, AALL0031 and COG studies, consistently stratify near-haploid and low-hypodiploid B-ALL subtypes as high-risk based on the poor prognosis of the patients, which does not depend on treatment era or on the NCI risk group in which they are classified [24,26,71]. In view of the poor prognosis of patients with hypodiploid B-ALL, they have been classically treated with high-dose chemotherapy followed by allogeneic transplantation. However, different studies assessing the impact of HSCT on B-ALL with near-haploidy and low-hypodiploidy failed to demonstrate a clear benefit of HSCT in MRD positive or negative patients [23,24,57,66] (Table 5). Notwithstanding these findings, the outcome of hypodiploid B-ALL with <40 chromosomes has been substantially improved by MRD-guided therapy, which intensifies treatments based on the MRD EOI status [23]. MRD-stratified treatment protocols are associated with a favorable outcome after adjusting for sex, age, and leukocyte count, with a 5-year EFS of 62% [24,72]. Interestingly, Jeha et al. described a 5-year EFS of 100% in a series of 6 patients with hypodiploidy of <44 chromosomes using MRD-guided therapy, and even in 1 patient with positive MRD EOI [72]. The favorable impact of treating patients according to protocols stratified by MRD EOI is likely related to the identification of chemosensitive patients, those that are MRD EOI negative, for whom HSCT does not represent any benefit, and to not undertreat MRD-positive patients. To evaluate the real impact of HSCT in this setting, trials comparing HSCT with chemotherapy alone in patients with positive MRD EOI would be necessary, but this would be difficult due to the low incidence and poor prognosis of these patients with only chemotherapy treatment [73]. Given the low survival in this group despite HSCT, it seems imperative to investigate novel treatments and therapeutic approaches. Interestingly, fit adult patients with hypodiploid B-ALL would likely benefit from pediatric-adapted more intensive chemotherapy schemes [74]; however, this is difficult to assess clinically due to the very low incidence and dismal prognosis of hypodiploid B-ALL adult patients.

### 7.4. Novel Therapeutic Targets and Approaches to Treat B-ALL with <40 Chromosomes

New treatments aiming to target recently identified biological drivers of hypodiploidies as well as immunotherapy strategies are currently being explored to achieve better responses before HSCT or to be used as alternative approaches. The recent discovery of near-universal *TP53* alterations in low-hypodiploid B-ALL has highlighted a key role for this gene in leukemogenesis. Investigation of this germinal mutation in this population is recommended when evaluating treatment with chemotherapy and HSCT. It remains to be demonstrated, however, whether therapies directed at this genetic lesion have an effect on low-hypodiploidy [58]. The anti-apoptotic protein BCL-2, has been identified as an effective therapeutic target for hypodiploid B-ALL with <40 chromosomes [75], and the efficacy of BCL-2 inhibitors (mainly venetoclax) has been demonstrated in ex vivo models of B-ALL with near-haploidy and low-hypodiploidy, especially in cases with elevated levels of the apoptosis-related factors BIM or BAD. Other authors have investigated the sensitivity of hypodiploid B-ALL cell lines and xenografts to MEK, PI3K, and the dual PI3K/mTOR inhibitors [30]. The PI3K and dual PI3K/mTOR inhibitors substantially inhibited proliferation of all tumors examined, suggesting that inhibition of the PI3K pathway may serve as a novel alternative therapy to treat hypodiploid B-ALL with <40 chromosomes [30].

Immunotherapy has shown encouraging results in relapsed/refractory B-ALL with high-risk cytogenetics, such as near-haploidy and low-hypodiploidy. Chimeric antigen receptor (CAR) T-cell and monoclonal antibodies or bi-specific T-cell engagers (BiTE), such as inotuzumab and blinatumomab, are the main immunotherapy approaches currently in use. It has been described that the patient response to CAR T-cell therapy does not depend on the cytogenetic risk group, with hypodiploid B-ALL representing up to 3.5% of cases according to the series [73]. Some of these cases have shown a promising response to CAR T-cell therapy, including an adult patient with low-hypodiploidy B-ALL in first relapse who received CD19-specific CAR T-cell therapy consolidated with HSCT [76]. Trials with CAR T-cell therapies as frontline approaches remain scarce and in some of them patients with hypodiploidy <40 chromosomes were excluded, such as the AALL1721/Cassiopeia trial (NCT03876769), a phase II single-arm trial of tisagenlecleucel (CD19-specific CAR T) in children and young adults with high risk B-ALL and persistent MRD at end of consolidation [77]. Blinatumomab, a BiTE antibody that directs T-cells to CD19-positive cells, has proven to be effective in adult patients with low-hypodiploid B-ALL, with 2 of the 4 patients reaching a complete response in one study [78]. It has also been used as an additional therapeutic approach in high-risk group protocols, which may induce a more in-depth response in these groups of B-ALL [79]. On the other hand, studies treating patients with inotuzumab ozogamicin, a humanized anti-CD22 antibody-drug conjugate, have not shown any specific high efficacy in patients with hypodiploid B-ALL [80]. Among the 3 patients with <40 chromosomes included in the study, 1 patient did not reach complete remission and the remaining patients did not reach a negative MRD EOI. In conclusion, new targeted treatments and immunotherapy approaches are very promising strategies, with ongoing protocols and active trials being conducted in patients with a very poor prognosis such as hypodiploidy ALL.

## 8. Concluding Remarks

Both near-haploid and low-hypodiploid B-ALL represent very rare entities, associated with a dismal clinical outcome. Such a low disease incidence represents a challenge to develop pre-clinical models aimed to study the etiology and pathogenesis of the disease and also represents a barrier to design statistically robust clinical trials.Cutting-edge cytogenetic and genetic assays must be implemented in routine diagnostic laboratories to distinguish between high-hyperdiploid and masked hypodiploid B-ALL patients since this has a major impact on patient treatment stratification and clinical outcome.The pathogenic effect(s) of chromosome losses and its contribution to leukemogenesis is currently not known. Furthermore, the biological contribution of chromosome doublings occurring in most of hypodiploid B-ALL cases with <40 chromosomes is poorly understood. Future studies aiming to decipher the biological mechanisms involved in the progression of hypodiploid B-ALL will help in the identification of innovative targeted treatments and/or diagnostic biomarkers to improve survival in these patients. 

## Figures and Tables

**Figure 1 cancers-14-00032-f001:**
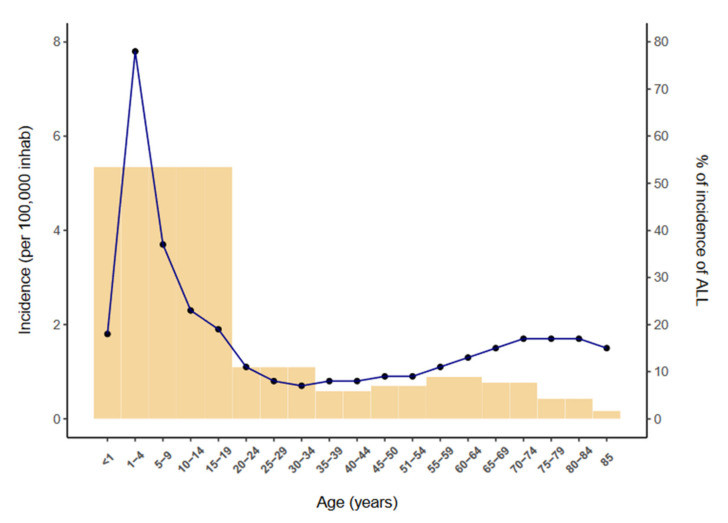
Incidence of ALL per 100,000 inhabitants by age (2014–2018) according to the SEER database [2].

**Figure 2 cancers-14-00032-f002:**
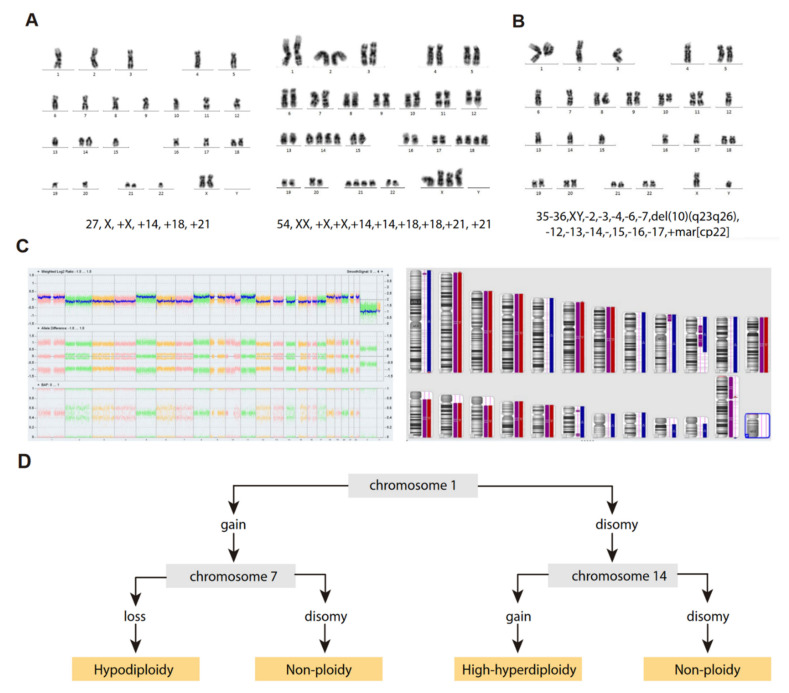
Cytogenetic characterization of B-ALL with <40 chromosomes. (**A**) G-banded karyotype of near-haploid B-ALL leukemic cells. *Left panel*, near-haploid clone. *Right panel*, chromosomally-doubled clone of the same patient. (**B**) G-banded karyotype of low-hypodiploid B-ALL leukemic cells. Karyotype formulas are indicated below. (**C**) SNP-array karyogram obtained for the low-hypodiploid B-ALL patient in B. *Right panel*, blue bars indicate chromosomal disomies of the duplicated/near-triploid clone, red bars indicate chromosomal losses, and purple bars indicate absence of heterozygosity. *Left panel*, Log2 ratio plot detailing whole chromosomal view for each chromosome, the figure demonstrates pattern of low-hypodiploidy where chromosomes with the lowest Log2 ratio represent the monosomies and a partial deletion of chromosome 10. Allele difference plot and B-allele frequency plot (BAF; BB, AB and AA alleles) indicates copy-neutral loss of heterozygosity. (**D**) Algorithm proposed by Creasey et al. [39] to distinguish hypodiploid with <40 chromosomes and high-hyperdiploid B-ALL cases based on specific chromosomal gains.

**Table 1 cancers-14-00032-t001:** WHO classification of precursor lymphoid neoplasms.

Precursor Lymphoid Neoplasms
B-lymphoblastic leukemia/lymphoma not otherwise specified (NOS)
B-lymphoblastic leukemia/lymphoma with recurrent genetic abnormalities B-lymphoblastic leukemia/lymphoma with t(9;22)(q34.1;q11.2); *BCR-ABL1* B-lymphoblastic leukemia/lymphoma with t(v;11q23.3); *KMT2A*-rearranged B-lymphoblastic leukemia/lymphoma with t(12;21)(p13.2;q22.1); *ETV6-RUNX1* B-lymphoblastic leukemia/lymphoma with hyperdiploidy B-lymphoblastic leukemia/lymphoma with hypodiploidy B-lymphoblastic leukemia/lymphoma with t(5;14)(q31.1;q32.1); *IGH-IL3* B-lymphoblastic leukemia/lymphoma with t(1;19)(q23;p13.3); *TCF3-PBX1* B-lymphoblastic leukemia/lymphoma, BCR-ABL1-like B-lymphoblastic leukemia/lymphoma with iAMP21
T-lymphoblastic leukemia/lymphoma Early T-cell precursor lymphoblastic leukemia
NK-lymphoblastic leukemia/lymphoma

**Table 2 cancers-14-00032-t002:** Clinical and biological features of patients with hypodiploid ALL.

Study	Near-Haploid								Low-Hypodiploid							
	M/F ^1^	Age		WBC (×10^9^/L) *				PreB/T Lineage	M/F ^1^	Age		WBC (×10^9^/L) *				PreB/T Lineage
		M ^2^	<10 yo at dx *	M ^2^	≤20	20–50	>50			M ^2^	<10 yo at dx *	M ^2^	≤20	20–50	>50	
SJCRH [16]	nr	4.5	3	nr				nr	nr	11	1	nr				nr
SJCRH-POG [17]	0.66	4.7	70	13	70	10	20	8/1^3^	2	10.5	22.2	8.4	66.6	11.1	22.2	5/1 ^3^
CCG [19]			63		50	13	38				nr		nr	nr	nr	
SJCRH [20]		7.3	100	2.5	100	0	0	4/0		nr	0	8.2	83.3	0	16.6	5/1
MRC [21]	0.75	7.4	64.3	67.3	38.5	30.8	30.8	13/0 ^3^	1.1	21.6	7.1	11	85.7	7.1	7.1	12/0 ^3,4^
Inc study [22]	1		69.6		39.1	41.3	19.6	nr	2.25		19.2		73	19.2	7.7	nr
SJTTS 15&16 [23]		3.6		8.1						13.9		5.6				
PDLWG [24]	1.46	6.2	73	21.8	49	20	31	99/1	1.14	12.9	26	7	85	10	4	99/1
Japanese study [25]	nr	nr	nr	nr	nr	nr	nr	3/0	nr	nr	nr	nr	nr	nr	nr	3/0

* Percentage of patients. ^1^ Male-to-female-ratio. ^2^ Median. ^3^ data not available from 4 patients in the SJCRH-PG study and in 1 patient in the MRC study. ^4^ one patient reported as “null” immunophenotype. Abbreviations: CCG, Children Cancer Group; CIBMTR, Centre for International Blood and Marrow Transplant Research; CNS, central nervous system; LH; low-hypodiploid; MRC, Medical Research Council; na, not applicable; NH, near-haploid; nr, not reported; PDLWG, Ponte di Legno childhood ALL Working group; POG, Pediatric Oncology Group; preB, B-cell precursor; SJCRH, Saint Jude Children’s Research Hospital; SJTTS, Saint Jude Total Therapy Study; WBC, white blood cell count at presentation.

**Table 3 cancers-14-00032-t003:** Cytogenetic characteristics of B-ALL patients with <40 chromosomes.

Age (Years)	Near-Haploid					Low-Hypodiploid					Reference
	MN	Retained chr	Lost chr	Doubled Clone	Frequency	MN	Retained chr	Lost chr	Doubled Clone	Frequency	
1–18	25–28	8, 10, 14, 18, 21 and sex chr.		yes	0.0046	30–40	1, 19, 21, 22 and sex chr.	3, 7, 13, 16, 17	yes	0.41	[17]
1–10	24–28	8, 10, 14, 18, 21 and sex chr.	7, 13, 14, 20, X	yes	0.0042	33–44		7, 13, 14, 20, X	nr	0.79	[19]
2–15	23–29	14, 18, 21 and sex chr.		yes	0.0039	33–39	1, 2, 5, 6, 8, 10, 11,12, 14, 18, 19, 21, 22 and the sex chr.	7, 17	yes	0.39	[21]
15–84	-	-	-	-	-	30–39	1, 5, 6, 8, 10, 11, 15, 18, 19, 21, 22, X, Y	3, 7, 15, 16, 17	66 to 78 chr	0.05	[31]
15–55	-	-	-	-	-	33–39	nr	nr	nr	0.0008	[21]
15–55/>55	<30				0.0016	32–39	1	2, 3, 7, 9, 13, 15, 16, 17, 20, 4	64–74	3.85%	[40]
1–9/>10	24–29	14, 18, 21 and sex chr.	nr	nr	nr	33–39	nrec	3, 7, 16, 17	nr	nr	[10]
<31	24–31	14, 18, 21 and sex chr.	nr	yes	0.008	32–39	1, 8, 10, 11, 18, 19, 21 and 22	nr	nr	0.0064	[41]

Abbreviations: chr, chromosomes; MN, modal numbers; nr, non-reported; nrec, non-recurrent.

**Table 4 cancers-14-00032-t004:** Molecular characteristics of hypodiploid <40 chromosomes B-ALL (Adapted from Holmfelt et al., 2013 [30]).

Genes	Cellular Pathway	Near-Haploid B-ALL			Low-Hypodiploid B-ALL		
		Mutation	Focal Deletion	Focal DEL + Mut	Mutation	Focal Deletion	Focal DEL + Mut
NF1	RTK/RAS pathway	11/68 (16%)	16/68 (24%)	3/68 (4%)	0	2/34 (6%)	0
KRAS		2/68 (3%)	0	0	0	0	0
NRAS		10/68 (15%)	0	0	0	0	0
PTPN11		1/68 (1%)	0	0	0	0	0
FLT3		6/68 (9%)	0	0	0	0	0
CRLF2		0	2/68 (3%) *	0	0	0	0
MAPK1		1/68 (1%)	0	0	0	0	0
GAB2		0	2/68 (3%)	0	0	1/34 (3%)	0
EPHA7		0	2/68 (3%)	0	0	0	0
RASA2		0	2/68 (3%)	0	0	0	0
IKZF1	B-cell development	0	3/68 (4%)	0	0	1/34 (3%)	0
IKZF2		1/68 (1%)	0	0	0	18/34 (53%)	0
IKZF3		1/68 (1%)	8/68 (12%)	0	0	1/34 (3%)	0
PAX5		1/68 (1%)	4/68 (6%)	0	0	2/34 (6%)	0
EBF1		0	0	0	0	0	0
VPREB1		0	3/68 (4%)	0	0	2/34 (6%)	0
CDKN2A/B	Cell cycle and apoptosis	0	15/68 (22%)	0	0	8/34 (24%)	0
TP53		2/68 (3%)	0	0	31/34 (91%)	0	0
RB1		2/68 (3%)	3/68 (4%)	1/68 (1%)	5/34 (15%)	8/34 (24%)	0
ETV6	Hematopoiesis	1/68 (1%)	3/68 (4%)	1/68 (1%)	0	0	0
Histone cluster (6p22)	Histone-related	0	13/68 (19%)	0	0	1/34 (3%)	0
ARID1B		0	2/68 (3%)	0	0	0	0
PAG1	BCR signalling	1/68 (1%)	6/68 (9%)	0	0	1/34 (3%)	0
ARPP21	Calmodulin signalling	0	1/68 (1%)	0	0	0	0
SLX4IP (C20orf194)	Telomere length maintenance	0	2/68 (3%)	0	0	0	0
CUL5	Ubiquitin pathway	0	2/68 (3%)	0	0	0	0
FAM53B	Wnt signalling	0	2/68 (3%)	0	0	0	0
PDS5B (APRIN)	Cohesin complex	0	2/68 (3%)	0	0	0	0
ANKRD11	Cell adhesion	0	0	0	0	2/34 (6%)	0
DMD		0	0	0	0	1/34 (3%)	0

* one patient encoding P2RY8-CRLF2.

**Table 5 cancers-14-00032-t005:** Event-free survival reported for patients with hypodiploid B-ALL by several relevant studies. Colored cells indicate statistical comparisons (blue, between chromosome numbers; orange, between other variables).

Study group	Years	Outcome variable	MN	Other variables	n	Mean (%)	*p*-value	Reference
Total Therapy Studies IX–XI Pediatric Oncology Group (POG)	1979–1988 1986-1988	3-year-DFS	<30 chr 30–40 chr 41–44 chr		109 8	40 33 37		[17]
Children’s Cancer Group (CCG)	1988–1995	6-year-EFS	33–34 chr 29–32 chr 24–28 chr		15	40		[19]
			30–40 chr		0	NA		
			41–44 chr		8	25		
Total Therapy protocols T11–T14	1984–1999	5-years EFS	36–44 chr			17		[20]
			25–29 chr			26		
			≥45 chr			75	<0.001	
			<45 chr			20		
Medical Research Council, UK ALL trial protocols	1990–2002	3-year-EFS	42–45 chr		121	65	0.0002	[21]
			25–39 chr		20	29		
AIEOP-3; BFM-5; CCG-33; COALL-3; DANA FARBER-4; POG-44; SJCRH-6; UK-20; NOPHO-6; and EORTC-15	1986–1996	8-year EFS	44 chr		50	52	<0.01	[22]
			40–43 chr		8	19		
			33–39 chr		26	37		
			30–32 chr		0	NA		
			24–29 chr		46	28		
COG AALL0031	2002–2006	4-year EFS	<30–44 chr	Chemotherapy alone	26	50	0.65	[66]
				Chemotherapy + BMT	13	62		
St. Jude Total Therapy Study XV&XVI	2000–2014	5-year EFS	24–31 chr		8	73	0.8	[23]
			32–39 chr		12	75		
				MRD EOI neg	14	85	0.03	
				MRD EOI pos	6	44		
CIBMTR-All BMT in CR1 or CR2	1990–2010	5-year-DFS	44–45 chr		39	64	0.01	[67]
			≤43 chr		39	37		
Children’s Healthcare of Atlanta	2004–2016	2-year DFS	32–39 chr		5	60	0.853	[57]
			24–31 chr		7	71		
				MRD EOI neg	10	69		
				MRD EOI pos	2	50		
				Age ≥ 10 years	6	33	0.021	
				Age < 10 years	6	100		
COG AALL03B1-COG AALL0331 and AALL0232 *	2003–2011	5-year EFS	>46 chr		NR	85	<0.01	[26]
			<44 chr		131	52		
				BMT en CR1	61	56	0.62	
				No BMT	52	49		
				MRD pos	30	26		
				MRD neg	74	64		
				Standard NCI risk group	48	60	0.026	
				High NCI risk group	83	47		
				Chemotherapy ^1^		4	0.13	
				HSCT ^1^		7		
Ponte di Legno Childhood ALL Working Group- ≤44 chromosomas	1997–2013	5-year EFS	44 chr		40	74	0.053	[24]
			40–43 chr		13	58		
			30–39 chr		118	50		
			24–29 chr		101	56		
				HCT in LH	21	64	0.89	
				No HCT in LH	93	62		
				HCT in NH	19	51	0.6	
				No HCT in NH	82	44		
TCCSG, JACLS, Japanese Children’s Cancer and Leukemia Study Group, Kyushu-Yamaguchi Children’s Cancer Study Group	1997–2012	5-year EFS	45 chr		101	73	<0.036	[25]
			44 chr		8	88		
			<44 chr		8	38		
				Relapse rate in patients < 44 chr	5	63		

* B-ALL patients 1–30-years-old. ^1^ The 5-year cumulative incidence of SMN. Abbreviations: chr, chromosomes; DFS, disease-free survival; EFS, event-free survival; HCT, hematopoietic stem cell transplant; LH, low-hypodiploidy; NH, near-haploidy.

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
