# Peer review of "Near-Haploidy and Low-Hypodiploidy in B-Cell Acute Lymphoblastic Leukemia: When Less Is Too Much"

_cancers, 2021, doi:10.3390/cancers14010032_

Round 1

Reviewer 1 Report

In this manuscript, the authors describe the characteristics of hypodiploid B-ALL subtypes focusing on near-haploid and low-hypodiploid B-ALL that represent extremely rare entities, associated with a very poor prognosis.

As pointed out by the authors throughout the text and in the conclusions, the paucity of the cases limits the development of preclinical models and of dedicated clinical trials. Therefore, the extensive review turns out to be mainly descriptive and in some parts of the text slightly repetitive.

The tables are confusing and difficult for the reader to consult (especially Table 2).

Author Response

We thank the reviewer’s for these helpful comments. Following reviewer’s advice, Table 2 has been reduced to easy readers’ consultation. We removed the “number of patients” columns, since they were not adding crucial information for the MS. In addition, the following changes were added throughout the MS to easy Table consultation: (i) we shaded in light-grey one every other row, and (ii) we rearranged the text throughout the MS to avoid Table splitting in two pages whenever possible.

Reviewer 2 Report

Molina et al. present a literature review on B-lymphoblastic leukemia/lymphoma with hypodiploidy, focusing on near-haploidy and low-hypodiploidy subgroups. The issue is well introduced, and arguments are appropriately presented.

First of all, the authors define the different hypodiploid B-ALL subgroups.

Later, they report main biological and molecular aspects and cytogenetic characterization of these entities, placing emphasis on the question of “masked hypodiploidy”.

Finally, the authors propose the principal theories about the origin of hypodiploidy and conclude displaying clinical outcome and treatment strategies available for these patients.

The work provides a useful overview of the long-established and emerging evidence on a rare WHO entity associated with a dismal clinical outcome, that imposes a routine diagnostics implementation and further studies to increase knowledge of its pathogenetic mechanisms.

Manuscript is well written and exhaustively referenced and results suitable for publication.

Author Response

We thank the reviewer for the thoughtful review of our MS.

Reviewer 3 Report

The authors present a very thorough, detailed and extensive review of a rare subtype of poor-ALL with hypodiploidy.   The manuscript is extremely well written with no language corrections required.  The limitations in knowledge (lack of confident best therapy for this group of patients) is well documented.  The paper provides a valuable review of a rare disease subgroup.

Author Response

(The authors gave the same response as above.)
